# On Ontology-Based Tourist Knowledge Representation and Recommendation

**Mao-Yuan Pai** [1,*] , **Ding-Chau Wang** [2] , **Tz-Heng Hsu** [3] , **Guan-Yu Lin** [4] and **Chao-Chun Chen** [5]

1    General Research Service Center, National Pingtung University of Science and Technology, Pingtung 912, Taiwan
2    Department of Information Management, Southern Taiwan University of Science and Technology, Tainan 710, Taiwan; dcwang@mail.stust.edu.tw
3    Department of Computer Science & Information Engineering, Southern Taiwan University of Science and Technology, Tainan 710, Taiwan; hsuth@mail.stust.edu.tw
4    Department of Computer Science & Information Engineering, National Cheng Kung University, Tainan 710, Taiwan; lgyprocoen@gmail.com
5    Institution of Manufacturing Information & Systems, Department of Computer Science & Information Engineering, National Cheng Kung University, Tainan 710, Taiwan; chaochun@mail.ncku.edu.tw
*    Correspondence: mypai@g4e.npust.edu.tw

**Abstract:** In the rapid development of the information technology age, many travelers search for travel articles through the Internet. These travel articles include the experience and knowledge of traveler, which can be used as a reference for tourism planning and attraction selection. At present, the most travel experience and knowledge is available in online travel reviews (OTR). OTR and eWOM (electronic word-of-mouth) contain a lot of knowledge of consumers and travelers. Many travelers often look for OTR content through virtual communities, blogs, and search engine, but the search results often cause information overload problems. In addition, through virtual communities, blogs, and search engines, an OTR search still requires using keywords. However, most travelers cannot know the name of the attraction; therefore, travelers cannot use the correct keywords to search. That causes travelers to be unable to get enough information from OTR and unable to make the best travel plan. Therefore, this study focuses on the ontology-based tourist knowledge representation and recommendation method. And the study is to search for popular attractions from the OTR content and construct a tourist knowledge structure for these travelers. When the tourists do not need to know the keywords of the popular attraction name, they just need to get their current location; and then ORT content will recommend the next attraction to the traveler, which helps the traveler make the correct travel decision. The evaluation result showed that the method proposed in this study can help the travelers to quickly make the travel decision and is better than the traditional searching methods.

**Keywords:** online travel reviews (OTR); electronic word of mouth (eWOM); ontology; travel recommendation; knowledge management

## 1. Introduction

Many travelers enjoy sharing their experiences or knowledge with others, and this can then be used as a reference by others when planning their trips, thus enabling them to learn more about the attractions they are going to visit [1]. Tourist knowledge is thus defined in this work as the opinions, experiences, and appraisals of travelers regarding target tourist attractions The most commonly used form of traveler knowledge is now online travel reviews (OTR), which are shared via virtual communities or blogs [2–4]. The value of tourist knowledge in OTR can be raised if it is first analyzed,

so that the positive appraisals of attractions can then be read by users. Nevertheless, there are some problems that knowledge engineers must address in order to analyze the data in OTR. First and foremost, OTR are usually posted on a variety of virtual communities or blogs and stored as webpage content rather than filed documents, making it difficult to extract, manage, analyze, and apply the tourist knowledge they contain.

Ontology is a methodology used to organize knowledge. Using classification and hierarchy, an ontology is able to store and express the knowledge in a certain field. A number of researchers have applied ontology to express tacit knowledge. For example, Chen and Chen applied it to analyze the discussions on web forums, in order to identify the evolution of discussion topics [5]. Pai et al. employed ontology to examine the knowledge structure of electronic word-of-mouth (eWOM) [6]. Such research primarily utilizes ontology to express tacit knowledge, and since OTR usually contains knowledge related to tourist opinions and experiences, it can be seen as this kind of knowledge. Therefore, OTR may also be expressed via an ontology. Similarly, Weng et al. applied an ontology to classify web content [7], while Lee et al. employed this approach to develop an automatic summarization mechanism for web news [8], and other studies have used ontology to develop a platform for semantic inquiry [9,10]. Sine OTR is a kind of web content, the ontology structure may also be applied to this, and thus this is the approach adopted in the current study.

The most common form of web search is based on keyword searches, but this can lead to the problem of information overload [11–14]. This means that travelers may need to browse a huge amount of content to find what they are looking for, which makes efficient tour planning and decision-making more difficult. In addition, when travelers do not know the names of popular attractions, they cannot use the correct keywords to search for attractions. If the user replaces keywords with their current location, virtual communities, blogs, and search engine can only provide the relevant content based on keyword; but cannot find the information about the other attraction. To address this issue, tourist businesses have developed a large number of package tours that they offer on the Internet [15], but these still require considerable time and effort to be compared in order to find the most suitable package [16–18]. Moreover, the package tours provided by tourist businesses may not satisfy consumer needs and are usually not flexible or subject to slight modifications. In addition, some travelers simply do not like to plan their tour in advance, but do so in real-time, while they are on their vacation. This is when a tourist attraction recommendation system can be especially useful, and there are currently many such systems. These can search for and recommend nearby attractions based on the travelers' location, as seen with the Touriseye mobile application. Nonetheless, these applications mostly rely on information shared by other users and do not provide access to information from other platforms, making comprehensive tourist information hard to obtain. Two problems faced by tourist information systems are thus how to offer sufficient information to travelers while also reducing the amount of time needed to examine it.

A number of researchers have examined tourist recommendation systems. For example, Yeh and Cheng (2015) used experts and the Delphi method to solicit tourist attraction information [19], but this approach required a lot of time and effort. To avoid this disadvantage, the current study primarily uses information technology to analyze OTR in a fast, efficient manner. In addition, Korfiatis and Poulos (2013) applied statistical analysis to analyze the information contained in an online platform, Booking.com, in order to recommend hotels [20]. However, the application of statistical analysis to tourist knowledge depends on the data being used, which may not be up-to-date. This study overcomes this problem by applying sentiment analysis to analyzing OTR content, and thus the latest appraisals of tourist attractions can be found. Among the current studies of tourist attraction recommendation systems [19–23], none has applied OTR and ontology.

The value of knowledge-based systems lies in the sharing and managing mechanism. Developing a brand new traveling recommendation mechanism is the primary purpose of the study. Through ontology, OTR was transformed into a traveler knowledge structure. The structure can help us to effectively carry out tourist knowledge management. In addition, the tourist knowledge structure

was also used to design a method for recommending tourist attractions. The method put tourist knowledge into effective sharing and use. Ontology has been a well-developed technique. Constructive ontology, deductive ontology, and natural language analysis were applied in the development of the recommendation mechanism. Nonetheless, current tourist attraction recommendation systems on the market have not used constructive ontology, deductive ontology, and natural language analysis. The study utilized well-developed technology to share and manage tourist knowledge, and this is what set this study apart from others. OTR is a kind of tacit knowledge. There are few tourist attraction recommendation systems that adopted tacit knowledge. The reason is that the collection, analysis, management, and application of tacit knowledge is more complicated. Tacit knowledge is adopted in this study. This has shown the value of this study in contributing to the application of tacit knowledge in this field.

Therefore, this study developed an ontology-based tourist knowledge representation and recommendation mechanism to carry out appraisal analysis of OTR content. Through the collection of positive appraisals related to tourist attractions, tourist knowledge structures were constructed based on ontology. These tourist knowledge structures were then used to develop a tourist knowledge representation and recommendation mechanism. This study thus has the following aims: (i) designing an OTR and eWOM management framework; (ii) constructing an ontology-based tourist knowledge representation and recommendation procedure; (iii) providing an ontology-based tourist knowledge representation and recommendation method; and (iv) developing an ontology-based tourist knowledge representation and recommendation mechanism.

This paper is organized as follows. In Section 2, the differences between OTR and ontology characteristics are analyzed, and the results are then used to develop an ontology-based tourist knowledge representation and recommendation model. In Section 3, the methodology used for designing and developing the ontology-based tourist knowledge representation and recommendation procedure is explained. In Section 4 the model is developed and tested to examine its practicality. Finally, the conclusions of this work are presented in Section 5, along with recommendations for future research.

## 2. OTR and Ontology Characteristics Analysis

In this section, the differences between OTR content and ontology characteristics are analyzed, and this is then used to develop an ontology-based tourist knowledge representation and commendation model.

### 2.1. Design of an OTR and eWOM Management Framework

eWOM is the user's feeling or experience after they use a product or service. Now, there are many types of eWOM. For example, many travelers often share online travel reviews (OTR) through social communities or blogs. OTR allow a traveler to express their own ideas and opinions about tourism, and eWOM shows the opinions and ideas of a production or service. Thus, ORT is a kind of eWOM. In order to effectively manage and analyze OTR content, the study will revise and propose a new application function of OTR and eWOM, which is based on Pai et al.'s (2013) [6] OTR (online travel reviews) and (eWOM) (electronic word-of-mouth) management framework, as shown in Figure 1.

(1).   eWOM collection: Since there is an overwhelming amount of eWOM available online, it is both difficult and inefficient to try and obtain all of it about a specific product or service. As a result, the eWOM collection process seeks to collect and filter eWOM-related content from blogs and web forums.

(2).   eWOM analysis: In order to understand the product or service appraisals contained in eWOM content, this study uses natural language processing and sentiment analysis to analyze it. The eWOM analysis can identify whether the product or service appraisals are positive or negative, and thus help eWOM knowledge representation and reasoning. The research method is eWOM collection and eWOM analysis, based on the Pai et al. (2013) eWOM analysis method [24].

(3).  Ontology-based SWOT analysis: The results of the eWOM analysis can be used to understand positive and negative appraisals of an enterprise as they will contain information about a firm's strengths, weaknesses, opportunities, and threats, thus helping strategic planning. Ontology-based SWOT analysis mainly uses computerized eWOM analysis techniques, the result of the related study will be reference to Pai et al. (2013) [6].

(4).  Consumer demands analysis: The content of eWOM and OTR contains a lot of opinions and appraisals from consumers and tourists. These opinions and appraisals can reflect the requirement of consumers and tourists. Therefore, we will develop eWOM and OTR appraisal as an analysis method of consumer's demand. The result of the related study will be reference to Lin et al. (2017) [25].

(5).  Knowledge representation and recommendation: OTR is a kind of eWOM, and from the eWOM analysis results, we can find out the customer's tourism evaluation through eWOM analysis. These appraisal are fragmented. Thus, knowledge representation and recommendation is mainly to construct consumer appraisal as a knowledge structure, and then make inference through the knowledge structure. OTR is a kind of eWOM. Thus, OTR can transform the consumer's travel experience into a tourism knowledge structure which help other travelers find more tourist attractions.

(6).  Appraisal-based BCG matrix analysis: The result of eWOM analysis can realize the positive and negative appraisal of an enterprise. This appraisal would include the strength, weaknesses, opportunities, and threats about an enterprise, the production, and the service. The function is mainly to identify the performance of the business and product line through the analysis technology of the automated eWOM. The eWOM analysis result will help the enterprise adjust their business or product and make the strategic planning efficient.

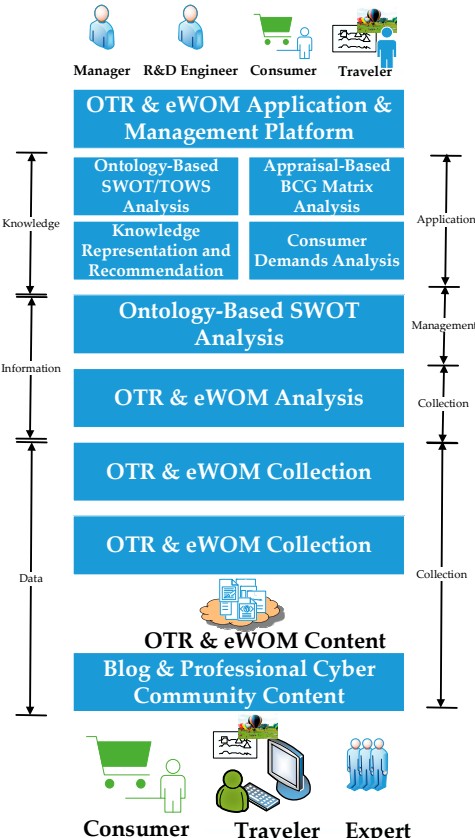

**Figure 1.** OTR (online travel reviews) and (eWOM) (electronic word-of-mouth).

As can be seen in Figure 1, the eWOM and OTR management framework is displayed. This study focuses on knowledge representation and recommendation, and the following section explains the procedures of knowledge representation and recommendation method development. The procedures, methods, and results are as follows:

### 2.2. Analysis of OTR Content Characteristics

In order to effective analyze OTR, this study first examined the characteristics of OTR content. The results were then used as the basis for developing the system. Related studies were reviewed to help organize the procedure for analyzing OTR content characteristics [5,6,24], as shown in Figure 2.

- Appraisal function: there are many positive and negative appraisals in OTR for a travel attraction. Thus, OTR appraisal can be used as a reference for filtering popular attractions.
- Popular attraction filtering: many studies show that online appraisals determine the user's willingness to purchase [6,24], which also means that the production with higher positive appraisals is the consumers' favorite. That is, the attraction gaining the higher positive appraisals will be tourists' preferences. Therefore, we can judge popular attractions through the total quantity of positive and negative appraisals.
- Popular attraction changing: consumer demand for products would change over time. [25] In the same way, travelers' evaluation of attractions will also alter with the time. When the requirement of consumers or travelers changes, the quantity of positive and negative appraisal also becomes different. Therefore, it is necessary to continuously collect the positive and negative appraisal of the attractions, which help travelers understand the latest tourist attractions.

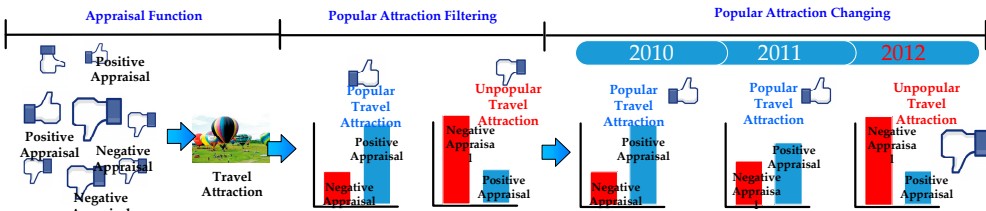

**Figure 2.** Analysis of OTR content characteristics.

This study applied set theory to explain the relationships between different tourist attraction sets. Each piece of OTR content contains a tourist attraction set. There are different types of sets among different tourist attraction sets, and they are classified into two types: disjoint sets and intersections, as explained below:

- Disjoint sets: If a specific tourist attraction cannot be found in two tourist attraction sets, it is likely that the two OTR do not overlap, and thus there is no direct relationship between the two tourist attraction sets.
- Intersection: If two tourist attraction sets share the same tourist attraction information, it means that the two OTR overlap. As shown in Figure 3, OTR 1 contains tourist attraction set 1, and the content includes K, L, M, N, O, P, Q, R, S, and T. As for OTR 2, it contains tourist attraction set 2, and the content has A, B, C, D, K, L, M, N, O, and P. OTR 1 and OTR 2 thus share some of the same tourist attraction information, namely K, L, M, N, O, and P. This indicates that author OTR1 and author OTR 2 may share similar appraisals regarding tourist attractions K, L, M, N, O, and P. In addition, OTR 1 content recommends not only "K, L, M, N, O, and P", but also "Q, R, S, and T". This means that there is a direct relationship among tourist attractions "K, L, M, N, and O" and "Q, R, S, and T". On the other hand, OTR 2 content recommends "K, L, M, N, O, and P" and "A, B, C, and D", which shows that there is a direct relationship among tourist attractions "K, L, M, N, O, and P" and "A, B, C, and D". Drawing on these relationships, when travelers want to

visit tourist attractions K, L, M, N, O, and P they can also be recommended to visit Q, R, S, and T, and A, B, C, and D.

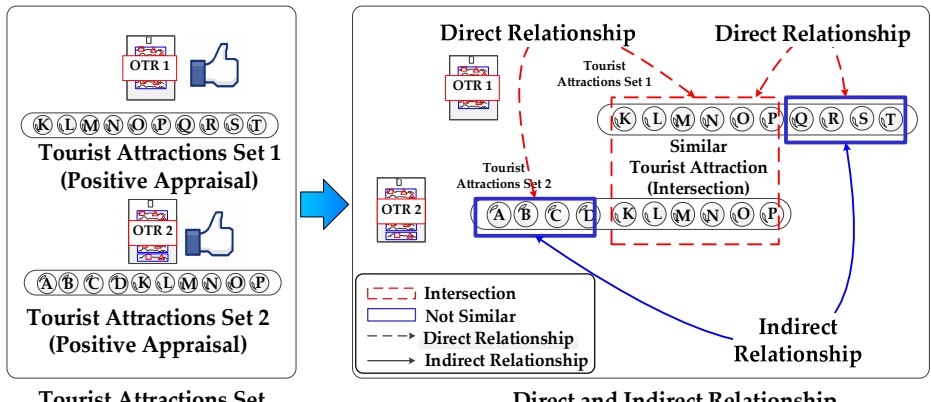

**Figure 3.** The relationships between tourist attraction sets.

Based on the notion that tourist attractions "K, L, M, N, O, and P" have direct relationships with "Q, R, S, and T" and "A, B, C, and D", there may be an indirect relationship between tourist attraction "Q, R, S, and T" (OTR1) and "A, B, C, and D" (OTR2). As seen in Figure 3, besides tourist attractions "K, L, M, N, O, and P", the authors of OTR1 and OTR2 content may also consider "Q, R, S, and T" and "A, B, C, and D" when they are recommending tourist attractions, When travelers want to visit tourist attractions "Q, R, S, and T", they may be recommended to visit tourist attractions "K, L, M, N, O, and P", due to the direct relationship. They may also be recommended to visit tourist attractions "A, B, C, and D" due to the indirect relationship.

There are thus latent relations between and among tourist attractions. In order to understand and effectively manage these, this study designed a tourist knowledge structure, as shown in Figure 4.

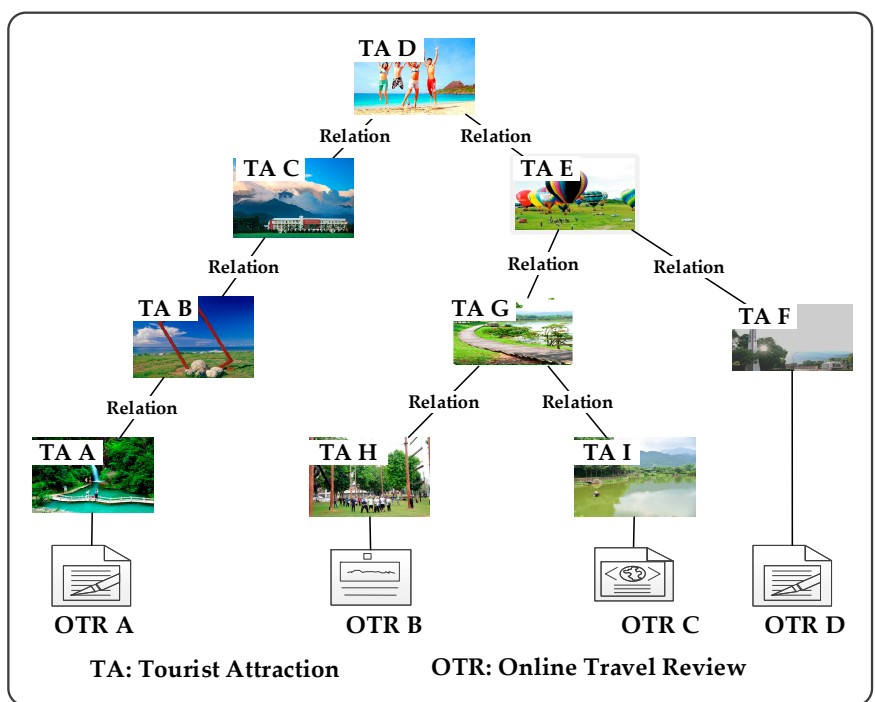

**Figure 4.** Tourist knowledge structure.

### 2.3. Design of the Tourist Knowledge Structure

In order to understand the tourist knowledge structure of OTR, the study used the research method of Chen and Chen (2012) [5], and Pai et al. (2013) [6], to explain the definition of "tourist knowledge structure relations" and "design of concept schema for tourist knowledge structure", as follows:

#### 2.3.1. Definition of Tourist Knowledge Structure Relations

Based on Figure 4, this section defines the tourist knowledge structure, based on the hierarchy relation, direct relation, and indirect relation. This is elaborated in Figure 5, and the detailed explanations are as follows:

- Hierarchy relation: A hierarchy relation indicates OTR content that includes tourist attraction information. For example, through hierarchy relation, it is expected to be possible to find tourist attractions related to OTR content.
- Direct relation: In addition to a hierarchy relation, there can also be a direct relation between attractions. There is a direct relation between a father node and a child node. As seen in Figure 5, tourist attraction C is the child node of tourist attraction D, which indicates that there is a direct relation between the two.
- Indirect relation: Since there is a direct relation between a grandfather node and a father node, and a direct relation also exists between a father node and a child node, it may be said that there is an indirect relation between a grandfather node and a child node. For example, tourist attraction C is the father node of tourist attraction B, and tourist attraction B is the father node of tourist attraction A. Tourist attraction C is the grandfather node of tourist attraction A. There is thus an indirect relation between tourist attraction C and tourist attraction A.

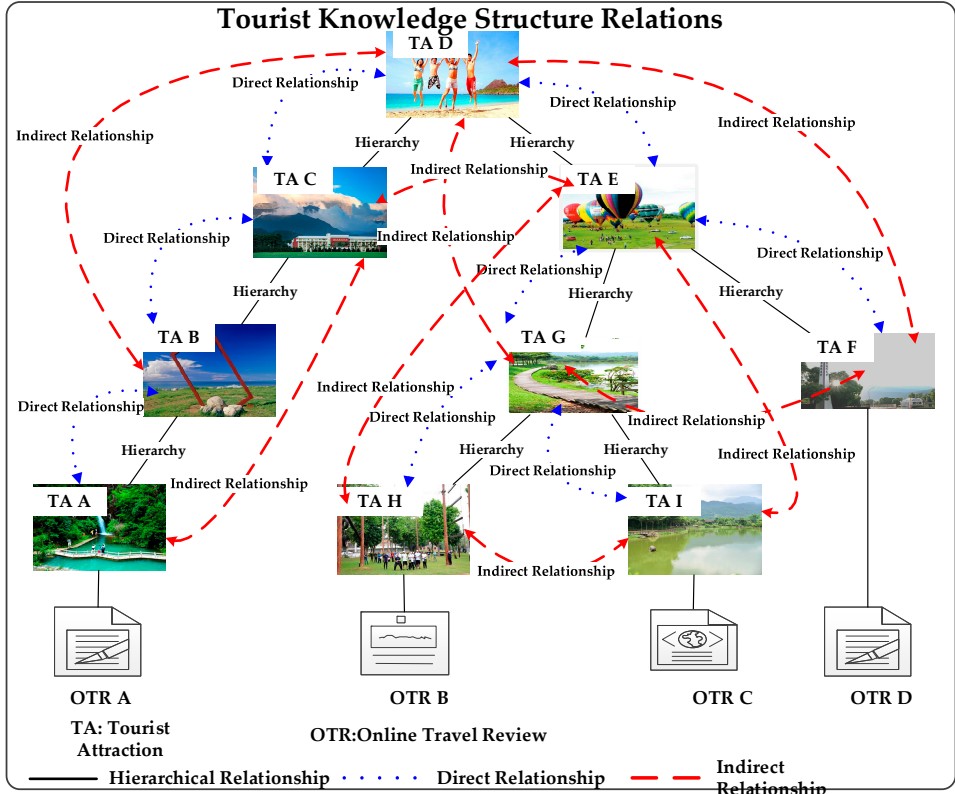

**Figure 5.** Relations between tourist attractions in the tourist knowledge structure.

2.3.2. Design of Concept Schema for the Tourist Knowledge Structure

Based on the ideas presented above, this study used an object oriented design model to design a schema for the tourist knowledge structure, as shown in Figure 6. This schema is composed of tourist attractions, attributes, and relations, and is explained as follows.

- Tourist attraction: Tourist attractions are the basic units that make up the tourist knowledge ontology, and this part of the schema records tourist attractions related to OTR content.
- Attribute: The tourist attraction attributes include (i) tourist attraction ID (number), (ii) tourist attraction name, (iii) tourist attraction district, (iv) tourist attraction review, and (v) distance. Tourist attraction district indicates the city or area in which the tourist attraction is located. Tourist attraction review indicates whether the tourist attraction has a positive or negative appraisal. Distance shows the distance between tourist attractions, as measured in kilometers.
- Relation: As indicated in the previous section, the relations among tourist attractions include the (i) hierarchy relation, (ii) direct relation, and (iii) indirect relation [26].

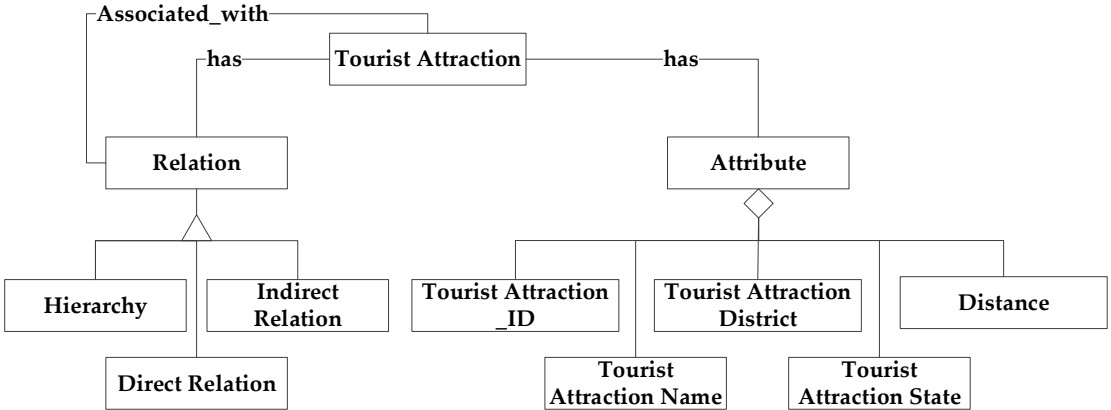

**Figure 6.** Tourist knowledge concept schema.

*2.4. Analysis of Ontological Characteristics*

Based on the results from Section 2.3, this study used ontology for the tourist knowledge structure, as shown in Figure 7. Ontology characteristics are analyzed as the basis to design the methods of recommendation for tourist attractions. Chen (2010) stated that in an ontology there is direct relation between a grandfather node and a father node [26], and a direct relation also exists between a father node and a child node, while there is an indirect relation between a grandfather node and a child node [26]. As can be seen in Figure 7, there is a direct relation between TA B and C, and a direct relation also exists between TA B and A, while there is an indirect relation between TA A and C. When travelers are located at TA G, the system will recommend attractions based on both direct and indirect relations. As a result, travelers can get four recommendations for tourist attractions: (1) TA E and D, (2) TA E and F, (3) TA H, and (4) TA I. Travelers are then able to select their own tourist attractions based on their preferences.

As shown in Figure 8, the tourist knowledge ontology includes hierarchical relations, and tour planning can be carried out using these. After the first and last destinations are chosen, the travelers can find related tourist attractions based on hierarchical relations. For instance, the first stop is set at TA C, and the last destination is set at TA I. D, E, and G are the tourist attractions related to TA C and I. Finally, the analytical results can be used to help travelers conduct their tour planning.

Ontological characteristics can thus be applied to tourist attraction reasoning and tour planning. On the basis of direct and indirect relations, tourist attraction reasoning helps people find popular tourist attractions. As for tour planning, the travelers first decide the first and last destinations, and then tourist attractions between these are recommended by the system.

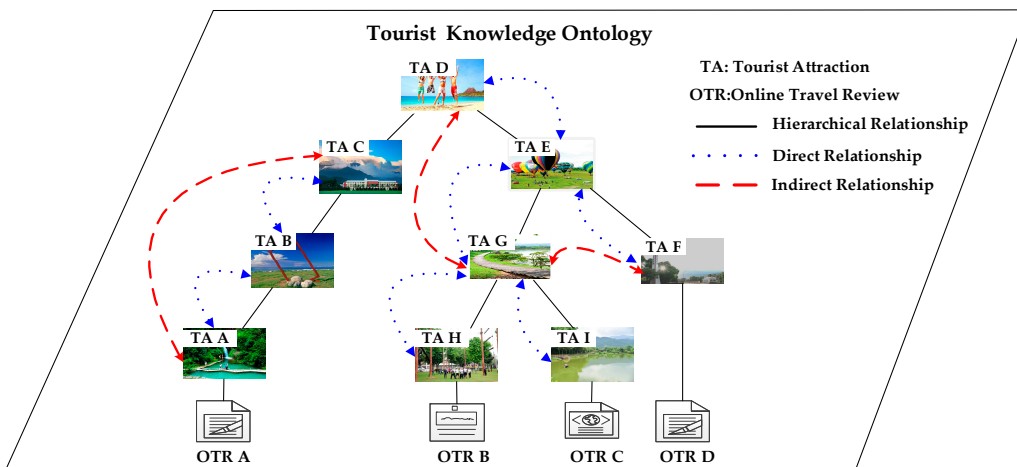

**Figure 7.** Tourist attraction recommendations with the tourist knowledge ontology.

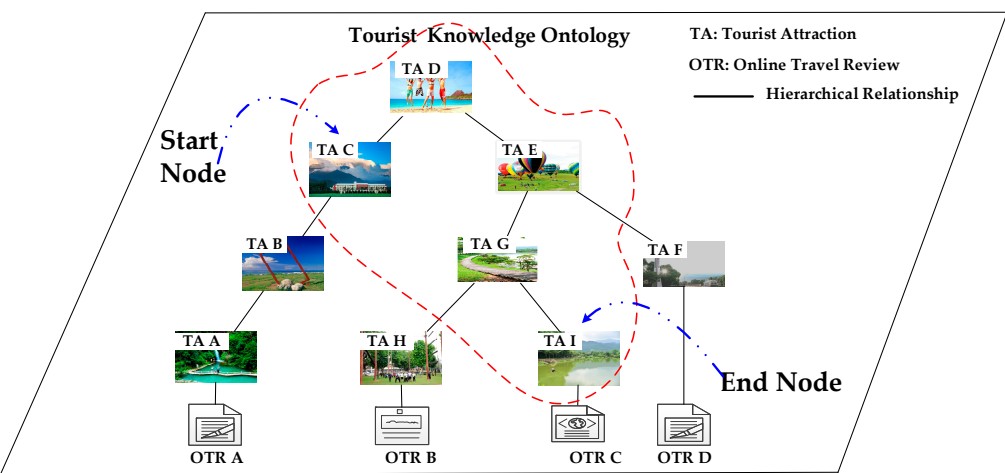

**Figure 8.** Tour planning with the tourist knowledge ontology.

## 3. Ontology-Based Tourist Knowledge Representation and Recommendation Method Development

Based on the OTR and ontology characteristics analysis results presented in Section 2, this study then designed an ontology-based tourist knowledge representation and recommendation procedure, which includes the following stages: (i) tourist attraction evaluation, (ii) tourist attraction ontology construction, and (iii) tourist attraction recommendation.

### 3.1. Procedure for Ontology-Based Tourist Knowledge Representation and Recommendation

Using the model presented in Section 2, this study designed a procedure for ontology-based tourist knowledge representation and recommendation, and this is expected to help travelers conduct their tour planning or tourist attraction selection. As shown in Figure 9, the procedure of ontology-based tourist attraction recommendation consists mainly of tourist attraction evaluation, tourist knowledge ontology construction, and tourist attraction recommendation. Tourist knowledge ontology construction involves concept set generation, hierarchy relationship generation, and distance calculation. Finally, tourist attraction recommendation involves tourist attraction matching, tourist attraction reasoning, tour planning, and tourist attraction sorting. These are discussed in more detail in the following section.

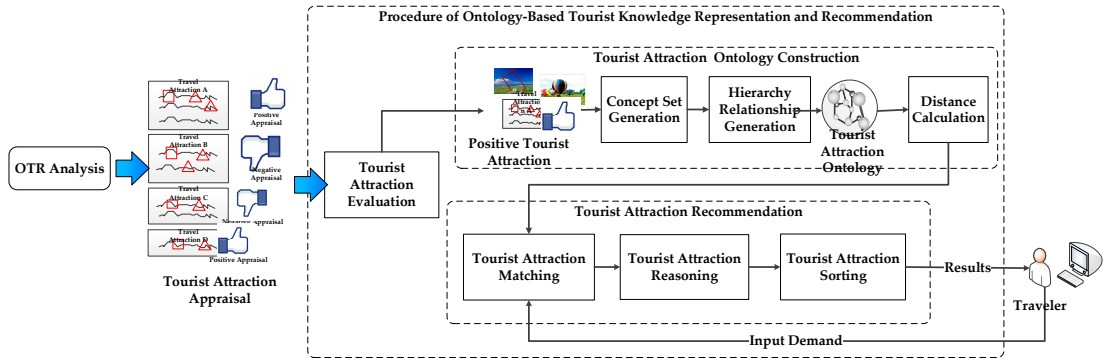

**Figure 9.** Procedure for ontology-based tourist attraction recommendation.

### 3.2. *Tourist* Attraction *Evaluation*

This study used Pai et al.'s (2013) research method to conduct OTR analysis [24]; the analysis result will be the source of tourist attraction evaluation. People usually prefer to visit tourist attractions that have positive appraisals. In order to see if an attraction has mainly positive or negative appraisals, this study adopted the majority rule as the basis for the evaluation. This method is carried out using Equations (1)–(3), which include the following variables: $POTR_k{}^p$ is the number of positive appraisals that the $k$th tourist attraction has in the tourist appraisal database. $p$ represents the positive appraisal set. $NOTR_k{}^n$ is the number of negative appraisals that the $k$th tourist attraction has in the tourist appraisal database. $n$ represents the negative appraisal set. $Allsum_k$ calculates whether tourist attraction $k$ has an overall positive or negative appraisal. If tourist attraction $k$ has a positive value, it means that it has a positive appraisal. In contrast, if tourist attraction $k$ has a negative value, this means that it has a negative appraisal.

$$POTR_k{}^p = \sum_{a=1}^{p} k_p \tag{1}$$

$$NOTR_k{}^n = \sum_{a=1}^{n} k_n \tag{2}$$

$$Allsum_k = POTR_k - NOTR_k \tag{3}$$

Based on Equations (1)–(3), this study designed a tourist attraction evaluation method, as shown in Figure 10, and described in more detail below.

Step 1.  Input the tourist attraction set, presented numerically as $k_i \in \{c_1, c_2, c_3, \ldots, c_j\}$, $C$ is the number of tourist attraction sets ($j = 1 \sim K$).

Step 2.  Calculate the number of negative appraisals that tourist attraction $k_i$ has in the database.

Step 3.  Calculate the number of positive appraisals that tourist attraction $k_i$ has in the database.

Step 4.  $Allsum_k$ calculates the overall appraisal of tourist attraction $k_i$. $C_k$ represents a temporary variable, and it stores the results of $Allsum_k$.

Step 5.  If the result of $C_k$ is a positive value, step 6 should be implemented. On the other hand, if $C_k$ is a negative value, step 7 should be implemented.

Step 6.  $C_k$ is a popular tourist attraction.

Step 7.  $C_k$ is not a popular tourist attraction.

Step 8.  Record the result that $C_k$ is a popular tourist attraction.

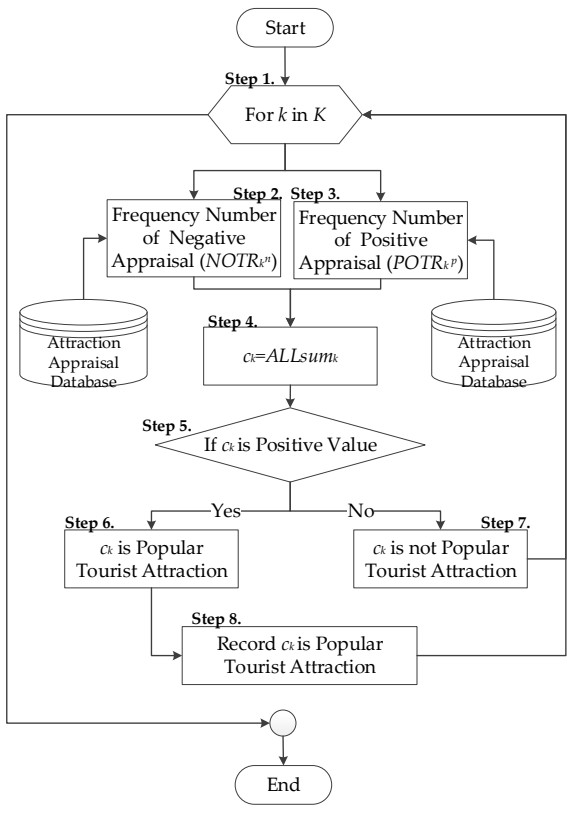

**Figure 10.** Process of tourist attraction evaluation.

This study used this algorithm and the related formula (Equations (1)–(3)) to calculate the results of the tourist attraction evaluation, as shown in Table 1.

**Table 1.** Results of the tourist attraction evaluation.

| Tourist Attraction | Number of Positive Appraisals (POTR) | Number of Negative Appraisals (NOTR) | ALL SUM | Evaluation Results |
|---|---|---|---|---|
| TA 1 | 57 | 91 | −34 | Negative tourist attraction |
| TA 2 | 107 | 50 | 57 | Positive tourist attraction |
| TA 3 | 49 | 189 | −140 | Negative tourist attraction |
| TA 4 | 54 | 22 | 32 | Positive tourist attraction |

### 3.3. Tourist Knowledge Ontology Construction

Formal concept analysis (FCA) is widely applied to knowledge structuring and organization in relation to online content [5–7,27,28], and thus it was used in the current study as the basis of developing a tourist knowledge ontology. The FCA is used to structure the hierarchical relationships among tourist attractions, based on the OTR content set and tourist attraction set. The main procedure of the tourist knowledge ontology construction includes the following steps: (i) concept set generation, (ii) hierarchy relationship generation, and (iii) distance calculation, as explained in more detail below.

### 3.3.1. Concept Set Generation

Based on the results reported in Section 3.2, the OTR content analysis, OTR content set ($y \in \{y_j, j = 1,2,3, \ldots ,N\}$)), and tourist attraction set that has positive appraisals can be found ($x \in \{x_i, i = 1,2,3, \ldots ,M\}$). The OTR content set and tourist attraction set were used to build a binary matrix, shown in Equation (4):

$$(x_M, y_N) = \begin{bmatrix} r(x_1, y_1) & r(x_1, y_2) & \cdots & r(x_1, y_N) \\ r(x_2, y_1) & r(a_2, y_2) & \cdots & r(x_2, y_N) \\ & & \vdots \vdots \ddots \vdots \\ r(x_M, y_1) & r(x_M, y_1) & \cdots & r(x_M, y_N) \end{bmatrix} \tag{4}$$

$r(a_m, b_n)$ indicates the relationship between the OTR content set and the tourist attraction set. If a tourist attraction appears in the OTR content set, then the value of $r(a_m, b_n)$ is 1, and otherwise the value of $r(a_m, b_n)$ is 0, as shown in Table 2.

**Table 2.** OTR and tourist attraction binary matrix.

|  | (x1) TA1 | (x2) TA2 | (x3) TA3 | (x4) TA4 | (x5) TA5 | (x6) TA6 | (x7) TA 7 | (x8) TA 8 | (x9) TA 9 |
|---|---|---|---|---|---|---|---|---|---|
| (y1) OTR A | 1 | 0 | 1 | 0 | 0 | 0 | 1 | 1 | 0 |
| (y2) OTR B | 0 | 0 | 1 | 0 | 1 | 0 | 0 | 0 | 0 |
| (y3) OTR C | 1 | 1 | 0 | 1 | 1 | 1 | 0 | 0 | 0 |
| (y4) OTR D | 0 | 0 | 0 | 1 | 1 | 1 | 0 | 1 | 1 |
| (y5) OTR E | 1 | 0 | 0 | 0 | 1 | 0 | 0 | 0 | 0 |
| (y6) OTR F | 0 | 1 | 0 | 1 | 0 | 0 | 1 | 0 | 1 |

$Y$ is defined as the subset of OTR content set $D$. The relationship between $Y$ and $D$ is marked as $Y \subseteq D$. $X$ is the subset of tourist attraction set $W$. The relationship between $X$ and $W$ is shown as $X \subseteq W$. If $X$ satisfies Equation (5) and $Y$ satisfies Equation (6), then $X$ and $Y$ is known as a concept $c$, which can be defined as $c(x, y)$, All of the concepts $c(x_i, y_j)$ are collectively defined as $C$. Based on Table 2, Equations (4)–(6) are used to carry out tourist attraction set generation, as shown in Figure 11.

$$\sigma(X) = \{w \in W \mid \forall d \in X : (w,d) \in I\} \tag{5}$$

$$\tau(Y) = \{d \in D \mid \forall w \in Y : (w,d) \in I\} \tag{6}$$

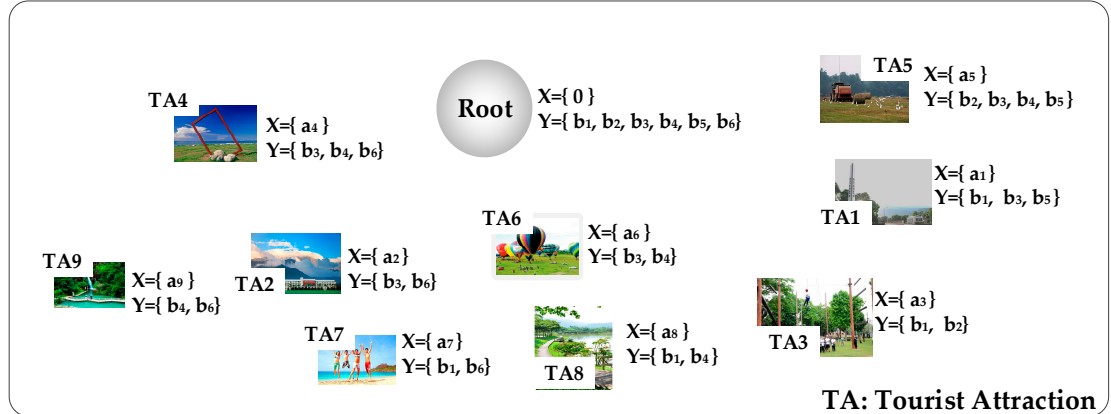

**Figure 11.** Tourist attraction set generation.

### 3.3.2. Hierarchy Relationship Generation

FCA obtains the concept hierarchy based on set operations related to mutual exclusiveness, intersections, and unions. Therefore, when tourist attractions $X_1$ and $X_2$ have similar OTR content and the former has more OTR content than the latter, this means tourist attraction $X_1$ is the super concept of

tourist attraction $X_2$. On the other hand, when tourist attractions $X_1$ and $X_2$ have similar OTR content and tourist attraction $X_1$ has less OTR content than tourist attraction $X_2$, this means tourist attraction $X_1$ is the child concept of tourist attraction $X_2$. This study thus applied set operations, as follows. If $X_1 \subseteq X_2$, $c_1(x_1,y_1)$ is the child concept of $c_2(x_2,y_2)$. The same tourist attraction may have many super concepts and child concepts to build the hierarchical relationships among tourist attractions. The study used Equation (7) to calculate the supremum of tourist attractions, and this can obtain the hierarchical relationships among tourist attraction words.

$$(x_1, y_1) \cup (x_2, y_2) = (\tau(y_1 \cap y_2), y_1 \cap y_2) \tag{7}$$

In addition to hierarchical relationships, tourist attraction*s* also have mutual relations with one another. Take tourist attractions $c_1(x_1,y_2)$ and $c_2(x_2,y_2)$ as an example; if $X_1 \subset X_2$ and $X_2 \subset X_1$, then $c_1$ and $c_2$ have a mutual relation. This study built the hierarchical relationships among tourist attractions based on Figure 11, as shown in Figure 12.

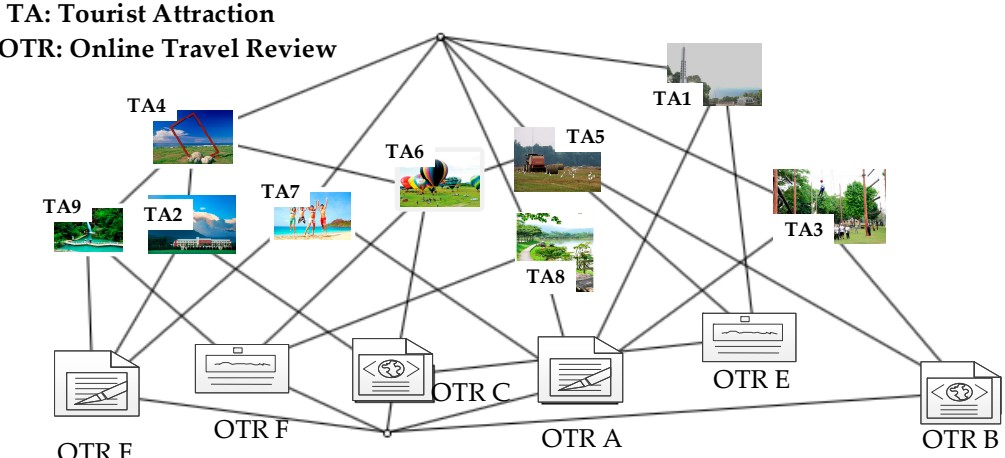

**Figure 12.** Hierarchical relationships among tourist attractions.

### 3.3.3. Distance Calculation

The distance calculation measures the distance between tourist attractions in the tourist knowledge ontology. In order to calculate this automatically, this study applied Google Maps API. The main procedures in the distance calculation are as follows: (i) obtain the tourist attraction input, (ii) use the HTML parser, and (iii) carry out distance extraction, as shown in Figure 13, explained as follows:

(1) Tourist attraction input: The names of tourist attractions from the tourist knowledge ontology were used as the data for Google Maps' API. As shown in Figure 13, if there is a relationship between tourist attractions A and B in tourist knowledge ontology, then they are found on Google Maps to obtain the distance between the two points.

(2) HTML parser: Every Google Maps platform has its own HTML tag format. In order to get the accurate distance between two tourist attractions, the HTML parser was used to analyze the HTML structure of the Google Maps' platform.

(3) Distance extraction: In this step, the HTML tags and web links were deleted, while the distance between tourist attractions was recorded and saved in the tourist knowledge ontology.

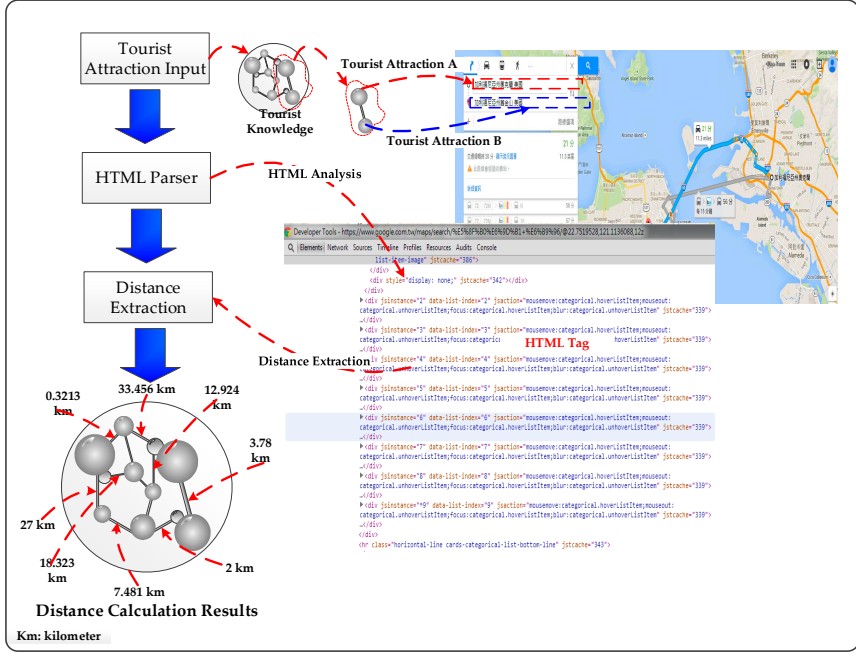

**Figure 13.** Process of distance calculation.

### 3.4. *Tourist* Attraction *Recommendation*

Based on the discussions presented in Section 3.3, this study designed a process of tourist attraction recommendation. The main procedure includes tourist attraction matching, tourist attraction reasoning, tour planning, and tourist attraction sorting, as explained below.

#### 3.4.1. Tourist Attraction Matching

When travelers are using tourist reasoning or tour planning, they usually search for the name of the target tourist attraction. Tourist attraction matching thus matches the attraction names from the travelers with those in the tourist knowledge ontology. In this way, the system may be able to find the tourist attraction information the travelers are looking for.

Tourist attraction matching applies the Jaccard Coefficient to make a comparison between tourist attraction names, with the calculation shown in Equation (8).

$$SubjectSimilarity(TA,\ TK_i) = \left| \frac{TA \cap TK_i}{TA \cup TK_i} \right| \tag{8}$$

The process of subject matching is based on Equation (8), as shown in Figure 14. "*TA*" represents the tourist attraction travelers are searching for. $TK_i$ is the $i$th tourist attraction name set from the tourist knowledge ontology (i.e., $TK_i = \{TK_1, TK_2, TK_3, \dots, TK_i\}$). $TS_j$ is a set of tourist attraction similarity values (i.e., $TS_j = \{TS_1, TS_2, TS_3, \dots, TS_j\}$).

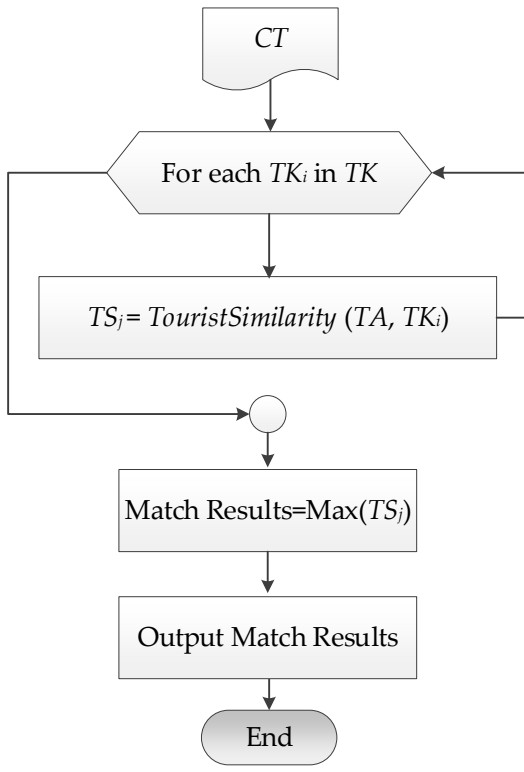

**Figure 14.** Process of subject matching.

### 3.4.2. Tourist Attraction Reasoning

The tourist attraction reasoning was based on Section 2.4., about the analysis of ontological characteristics. The location of travelers is used to find the direct and indirect relations in the tourist attraction reasoning. This process is explained below.

The tourist knowledge ontology is transformed into a relation matrix. $(X_i, X_j)$ represents the relationship between two tourist attractions. If the value of $(X_i, X_j)$ is 1, it means there is a relationship between tourist attractions $X_i$ and $X_j$. If the value of $(X_i, X_j)$ is 0, it means there is no relationship between tourist attractions $X_i$ and $X_j$, as shown in Figure 15.

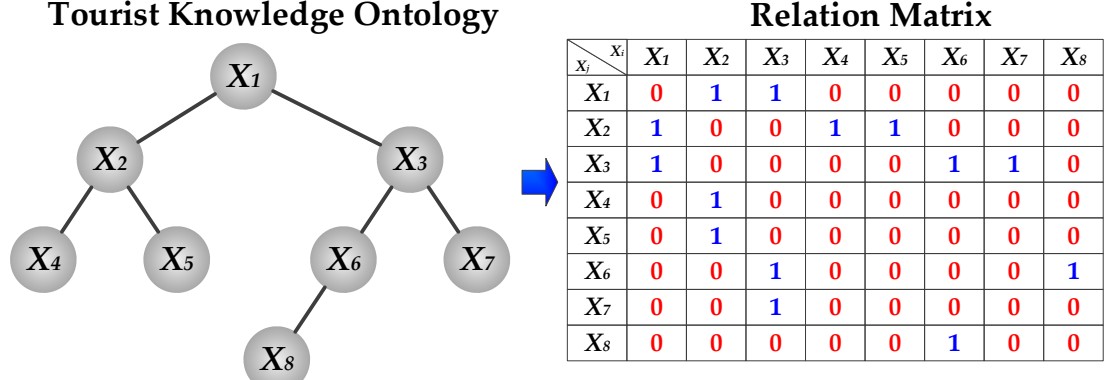

**Figure 15.** Tourist knowledge ontology transformation into a relation matrix.

As can be seen in Figure 15, tourist attraction $X_1$ is set as the location for the traveler. Tourist attraction reasoning was then carried out to see the direct and indirect relations among the attractions. The results of the reasoning are $(X_2, X_4)$, $(X_2, X_4)$, $(X_3, X_6)$, and $(X_3, X_7)$, respectively, as shown in Table 3.

**Table 3.** Record the results of tourist attraction reasoning.

| Steps | Seat (S) | Neighbor Node (N) | Action |
|:---:|:---:|:---:|:---:|
| 1 | $X_1$ | $X_1 \rightarrow X_2$, $X_1 \rightarrow X_3$ | The sub nodes of $X_2$ and $X_3$ are searched |
| 2 | $X_2$ | $X_2 \rightarrow X_4$, $X_2 \rightarrow X_4$ | Output the search results: $(X_2, X_4)$ and $(X_2, X_5)$ |
| 3 | $X_3$ | $X_3 \rightarrow X_6$, $X_3 \rightarrow X_7$ | Output the search results: $(X_3, X_6)$ and $(X_3, X_7)$ |

A computer program was used to automatically carry out tourist attraction reasoning, based on the algorithm shown in Algorithm 1.

---

**Algorithm 1.** Algorithm for tourist attraction reasoning.

---

Tourist knowledge ontology transformation into relation matrix $[X_i]$ $[X_j]$

Input relation matrix $[X_i]$ $[X_j]$//$X_i$ and $X_j$ represent the tourist attraction names of tourist attraction set, $X_i$ = $\{X_1, X_2, X_3, \ldots, X_i\}$, $X_j$ = $\{X_1, X_2, X_3, \ldots, X_j\}$.

Input present position $(L)$//$L$ represents the location of the traveler.

create queue$(Q)$//queue $(Q)$ is used to save the search results//

main(){

    **for** $X_i$ = 1 to $X_A$//The tourist attractions *of* $X_1$, $X_1$, … , $X_A$ are set as nodes that travelers have not yet visited, $X_A$

    //$X_A$ represents the number of tourist attraction sets

        visit$[X_i]$ = false; //The node $X_i$ is set as not visited.

    **for-end**

visit$[L]$ = true; //The current location of travelers, $(L)$, is set as visited

    **for** $X_i$ =1 to $X_A$//

        **If** (!visit$[X_i]$)//It searched the not visited node, $X_i$.

          visit$[X_i]$ = true; //The found node, $X_i$, is set as visited and will not be searched later.

          enqueue$(Q, X_i)$; //The found node, $X_i$, is saved to queue$(Q)$

      **if-end**

      **if** (all visit$[X_i]$)//It represents all the nodes $(X_j)$ has been visited.

        Traversal (front$(Q)$); //It requests traversal() function and loads the first information (node) from queue$(Q)$.

      **if-end**

    **for-end**

}

traversal $(X_i)${

  **while** (!empty(Q))//it judges if queue is null

    **for** $X_j$ = 1 to$X_A$

        **if** (relation matrix $[X_i]$ $[X_j]$ = 1 &&!visit$[X_j]$)//In relation matrix, the value of $(X_i, X_j)$ is 1, and noted $X_j$ is not visited. Then, there is a relationship between $X_i$ and $X_j$, and $X_j$ has not been searched by the program.

          visit$[X_j]$ = true; //The found node $X_j$ is set as visited and will not be searched later.

      *output* $(X_i, X_j)$; //Out put the result node, $X_j$

        traversal$(X_i)$; //It requests traversal() function and loads the value of node$(X_i)$

      **if-end**

      **if** (all visit$[X_j]$)//If all the nodes $(X_j)$ has been visited.

        dequeue$(Q)$; //It delete the front node of queue.

        traversal(front$(Q)$); //It requests traversal() function and loads the first information (node) from queue.

      **if-end**

    **for-end**

  **while-end**

}

---

### 3.4.3. Tour Planning

Based on the starting point and destination set by the travelers and tourist knowledge ontology, the tourist attractions between the starting point and destination are found. The method for tour planning is explained as follows.

In Figure 15, the tourist attraction $X_1$ is set as the starting point of a tour and $X_8$ is set as the final destination. Using the tourist knowledge ontology, tourist attractions related to $X_1$ and $X_8$ are found. The results of the tour planning are $X_1$, $X_3$, $X_6$, and $X_8$, as shown in Figure 16.

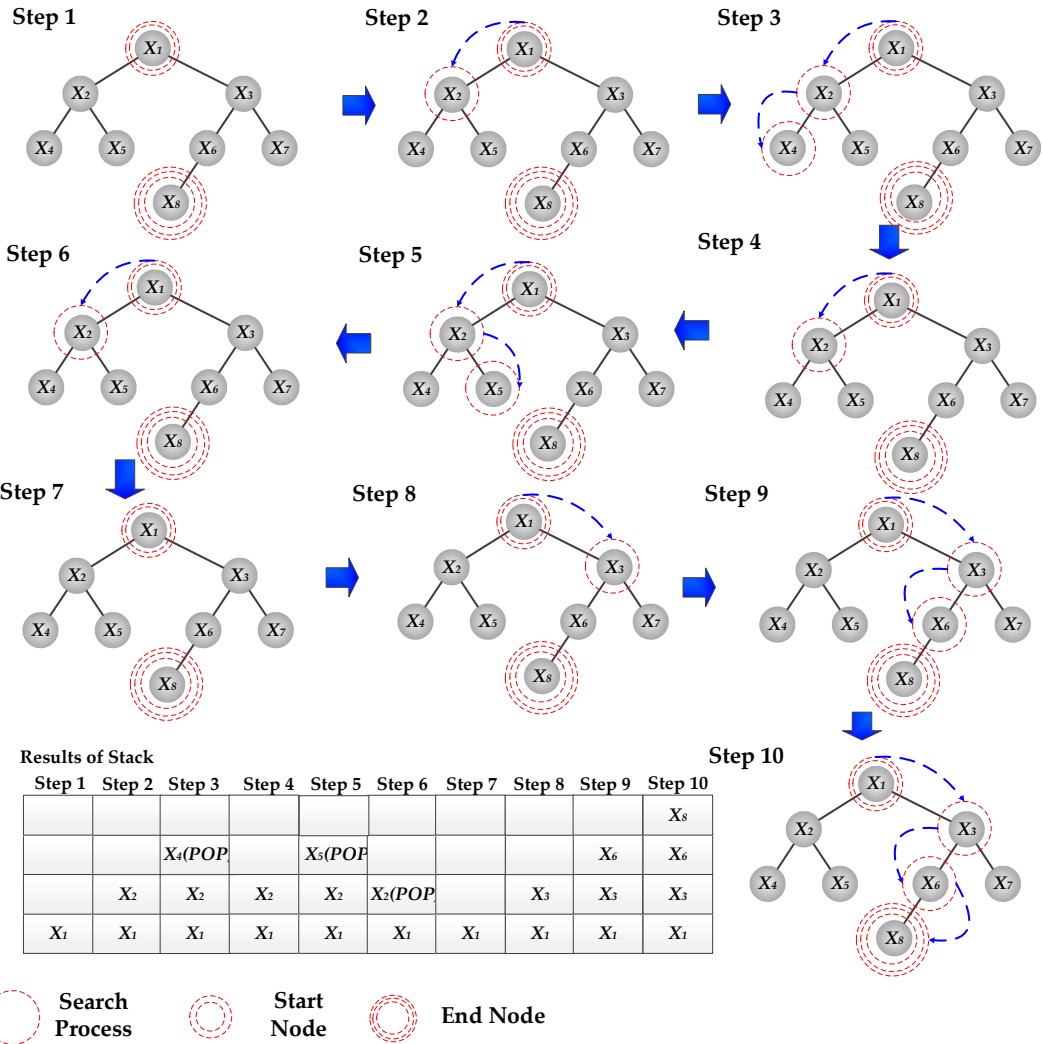

**Figure 16.** Process for tour planning.

In order to automatize the tour planning process, the study developed a tour planning algorithm, as detailed in Algorithm 2, below.

---

**Algorithm 2.** Algorithm for tour planning.

---

Tourist knowledge ontology transformation into a relation matrix $[X_i]$ $[X_j]$

Input relation matrix $[X_i]$ $[X_j]$

input start node $(X_s)$//The user inputs the start node $(X_s)$, $X_s$ is a node in the relation matrix.

input end node $(X_e)$//The user inputs end node $(X_e)$, $X_e$ is a node in the tourist knowledge ontology.

create stack(S)//It sets up stack.

main(){

    *for* $X_i = 1$ to$X_A$ //$X_1$ to $X_A$ are the nodes in the tourist knowledge ontology. The nodes from $X_1$ to $X_A$ are set as not visited.

        visit[$X_i$] = false;

    *for-end*

visit [$X_s$] = true; //The starting point $(X_s)$ is set as visited.

push $(S, X_s)$; //The starting point $(X_s)$ is put into the stack.

    *for* $X_i = 1$ to$X_A$

        *if* (!visit[$X_i$])//It searches the nodes that has not been visited $(X_i)$

          visit [$X_i$] = true; // It finds the node $(X_i)$ and sets it as visited.

          push $(S, X_i)$; //The node $(X_i)$ is put into the stack

          traversal $(X_i)$; //It requests traversal() function and loads node $(X_i)$

        *if-end*

    *for-end*

}

traversal($X_i$){

    *for*$X_j = 1$ to$X_A$

        *if* (relation matrix $[X_i]$ $[X_j]$ = 1 &&!visit[$X_j$])//The value of the relation matrix $(X_i, X_j)$ is 1, and the node $X_j$ is set as not visited.

            *if* ($[X_j]$ == $[X_e]$)//Node $(X_j)$ equals to destination node $(X_e)$

              output stack(S); //It outputs all the nodes in the stack.

            **else**//Node $(X_j)$ is different from destination node $(X_e)$

             visit [$X_j$] = true; //Node $(X_j)$ is set as visited.

              Push $(S, X_j)$; //Node $(X_j)$ is saved into the stack.

             traversal($X_j$); //It requests formula, traversal(), and loads node $(X_j)$

          *if-end*

          *if* (all visit[$X_j$])//All the nodes $(X_j)$ has been visited.

            pop(S); //The nodes of stack (pop) is deleted.

            traversal(top(S)); //It requests traversal() function and loads the first node (top(S)) on top of the stack.

          *if-end*

    *for-end*

}

---

### 3.4.4. Tourist Attraction Sorting

Based on Section 3.4.2, attraction reasoning, and Section 3.4.3, tour planning, there may be one or more than one results. The tourist attraction sorting ranks all the analytical results based on their distances. The nearest destination is ranked first and will be recommended to travelers. The method for tourist attraction sorting is explained as follows:

Section 3.3.3 showed how to measure the distance between tourist attractions $X_i$ and $X_j$. The relation matrix thus becomes a distance matrix. $(X_i, X_j)$ is used to represent the distance between tourist attractions, as displayed in Figure 17.

## Relation Matrix

| $X_j$ \ $X_i$ | $X_1$ | $X_2$ | $X_3$ | $X_4$ | $X_5$ | $X_6$ | $X_7$ | $X_8$ |
|---|---|---|---|---|---|---|---|---|
| $X_1$ | 0 | 1 | 1 | 0 | 0 | 0 | 0 | 0 |
| $X_2$ | 1 | 0 | 0 | 1 | 1 | 0 | 0 | 0 |
| $X_3$ | 1 | 0 | 0 | 0 | 0 | 1 | 1 | 0 |
| $X_4$ | 0 | 1 | 0 | 0 | 0 | 0 | 0 | 0 |
| $X_5$ | 0 | 1 | 0 | 0 | 0 | 0 | 0 | 0 |
| $X_6$ | 0 | 0 | 1 | 0 | 0 | 0 | 0 | 1 |
| $X_7$ | 0 | 0 | 1 | 0 | 0 | 0 | 0 | 0 |
| $X_8$ | 0 | 0 | 0 | 0 | 0 | 1 | 0 | 0 |

## Distance Matrix

| $X_j$ \ $X_i$ | $X_1$ | $X_2$ | $X_3$ | $X_4$ | $X_5$ | $X_6$ | $X_7$ | $X_8$ |
|---|---|---|---|---|---|---|---|---|
| $X_1$ | 0 | 13 | 2 | 0 | 0 | 0 | 0 | 0 |
| $X_2$ | 13 | 0 | 0 | 4 | 8 | 0 | 0 | 0 |
| $X_3$ | 2 | 0 | 0 | 0 | 0 | 64 | 23 | 0 |
| $X_4$ | 0 | 4 | 0 | 0 | 0 | 0 | 0 | 0 |
| $X_5$ | 0 | 8 | 0 | 0 | 0 | 0 | 0 | 0 |
| $X_6$ | 0 | 0 | 64 | 0 | 0 | 0 | 0 | 7 |
| $X_7$ | 0 | 0 | 23 | 0 | 0 | 0 | 0 | 0 |
| $X_8$ | 0 | 0 | 0 | 0 | 0 | 7 | 0 | 0 |

**Figure 17.** Relation matrix transformation into a distance matrix.

As Algorithm 2 showed, the results of tourist attraction reasoning are $(X_2, X_4)$, $(X_2, X_5)$, $(X_3, X_6)$, and $(X_3, X_7)$. The distance matrix reveals the distance between points. The distance of $(X_2, X_4)$ is 4, the distance of $(X_2, X_5)$ is 8, the distance of $(X_3, X_6)$ is 64, and the distance of $(X_3, X_7)$ is 23. These are then ranked by distance, and the one with the nearest distance is ranked first. The first recommended result is thus $(X_2, X_4)$, followed by $(X_2, X_5)$, $(X_3, X_7)$, and $(X_3, X_6)$. In order to automatize the tourist attraction sorting, this study developed an algorithm for tourist attraction sorting, as shown in Algorithm 3.

---

**Algorithm 3.** Algorithm for tourist attraction sorting.

---

input DistanceMatrix $[X_i]$ $[X_j]$//The users input the Distance Matrix, $X_i$ represents the tourist attraction name in the X axis. $X_j$ represents the tourist attraction name in the $Y$ axis.
main(){
   **while** $(X_i! = null\&\&X_j!= null)$//When the value of $(X_i, X_j)$ does not equal to 0 in the Distance Matrix,
      *attractionList*. Add (distanceMatrix$[X_i][X_j]$); //add $X_i$ and $X_j$ to the *attractionList*.
   **while-end**//When the value of $(X_i, X_j)$ equals 0 in the Distance Matrix, the process is terminated.
   sorting (*attractionList*); //Sorting function loads the value of attractionList.
}
sorting (attractionList){
  *for* $m = 0$ to (*attractionList.length* $- 1$)
  swap = false; //If there is no change of location, it is marked as false.
    *for* $n = 0$ to(*attractionList.length* $- 1$) $- m$
     **if** (*attractionList* $[n] >$ *attractionList* $[n+1]$)//If the value of current location is bigger than the value of the next location.
      temp = *attractionList* $[n]$; it executes the swap program, as explained above.
     *attractionList* $[n] = $ *attractionList* $[n+1]$;
     *attractionList* $[n+1] = $ temp;
     swap= true; //If there is a change of location, it is marked as true.
    **if-end**
   **for-end**
   **if**(swap == false) //There is no change of location
    break; //Exit for loop
   **if-end**
  **for-end**
outputattractionList//It outputs all the values in attractionList.
}

---

## 4. Prototype Implementation and System Evaluation

Based on the proposed tourist knowledge structure-based tourist attraction recommendation method, this section describes a prototype for implementing an analysis of the tourist attraction

recommendation system using C#. The technology used for this implementation, as well as the results with an illustrative case study, are described below.

*4.1. Implementation Environment*

The technology used for implementing the prototype include an Intel Core$^{TM}$ i5-760- 2.8GHz PC, Microsoft Windows 7 Professional, Internet Information Services (IIS), Microsoft SQL Server 2012, and Microsoft Visual Studio 2010. Figure 18 shows the tourist knowledge structure-based tourist attraction recommendation mechanism framework, which includes three layers of user interface, mechanism operation, and tourist knowledge repository, as explained below.

(1) User interface layer: The users of the program are travelers. The system provides two major functions: tourist attraction reasoning and tour planning. With regard to the former, after the travelers input their location, the program uses the direct and indirect relations to recommend nearby tourist attraction. For tour planning, the travelers first decide the starting and final tourist attractions. The program then uses the tourist knowledge ontology to find attractions between these two points.

(2) Mechanism operation layer: There are five modules in this mechanism, as follows: application module, tourist attraction evaluation module, tourist knowledge ontology construction module, and tourist attraction recommendation module.

- Application module: This includes the tourist reasoning application component, tour planning application component, and tourist knowledge structure loading component.
- Tourist attraction evaluation module: The analysis result will be the source of tourist attraction evaluation.
- Tourist knowledge ontology construction module: This includes a concept set generation component, hierarchy relationship generation component, and distance calculation component.
- Tourist attraction recommendation module: This includes the tourist attraction matching component, tourist attraction reasoning component, tour planning component, and tourist attraction sorting component.

(3) Tourist knowledge repository layer: This includes the following five repositories: OTR content repository, tourist attraction name repository, positive appraisal word repository, negative appraisal word repository, and tourist knowledge ontology (OWL) repository. The OTR content repository stores OTR content. The tourist attraction name repository stores tourist attraction names. The positive appraisal word repository stores positive appraisal words related to tourist attractions, while the negative appraisal word repository stores negative words. The tourist knowledge ontology (OWL) repository stores the relationships and attributes among nodes in the tourist knowledge ontology.

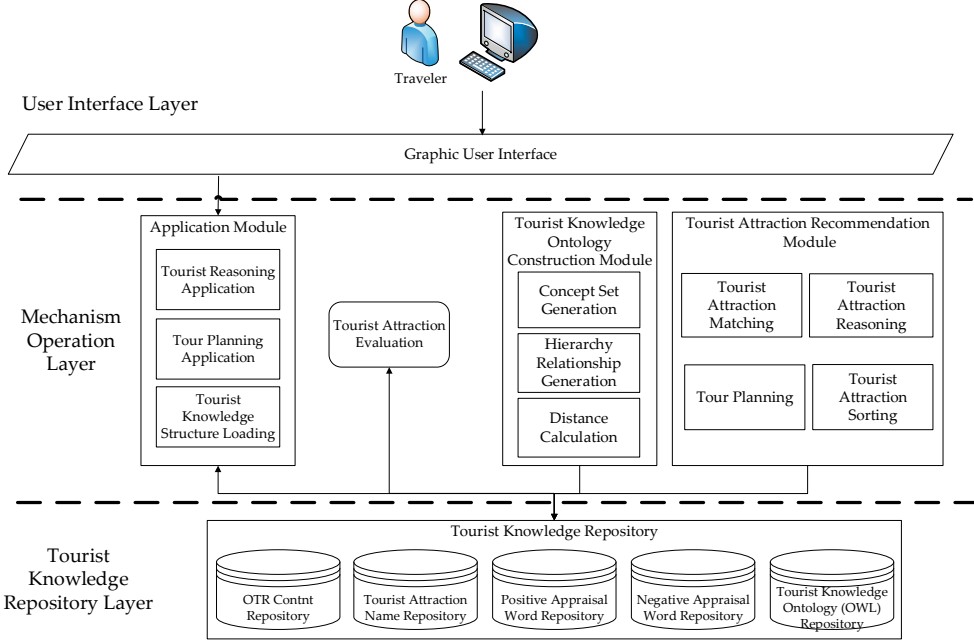

**Figure 18.** Ontology-based tourist knowledge representation and recommendation mechanism framework.

## 4.2. Implementation Results

Through the research of Pai et al. (2013) [24], the system conducted OTR collection and appraisal analysis. The OTR source is from Taiwan PTT (Taiwan bulletin board system name) Forum, of which the content is related to Southern Taiwan (Pingtung County, Kaohsiung City, Tainan City, and Chiayi County). The study collated 9146 of OTR content found 720 attractions through the research method. Therefore, 720 attractions were constructed and identified as tourist knowledge. Finally, this study will be as a system, shown as Figures 19–23. Figure 19 shows the selection of the recommendation model. Figure 20 shows the input of the traveler's location, while Figure 21 shows the results of tourist attraction reasoning model. Figure 22 shows the input start and end nodes, and Figure 23 shows the results of the tour planning model.

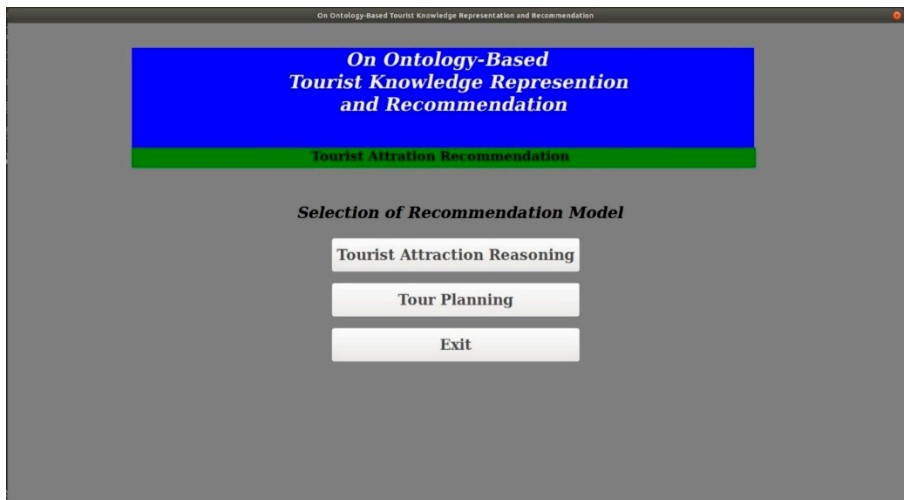

**Figure 19.** Selection of recommendation model.

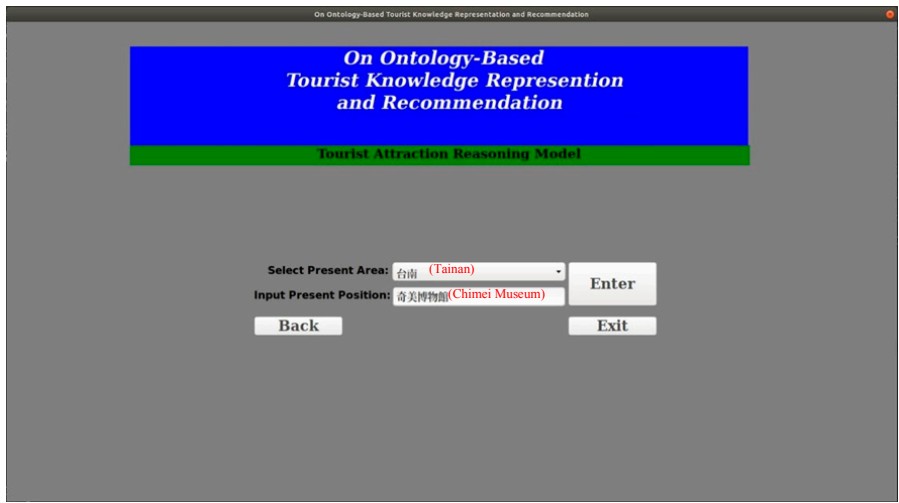

**Figure 20.** Input the location of the traveler.

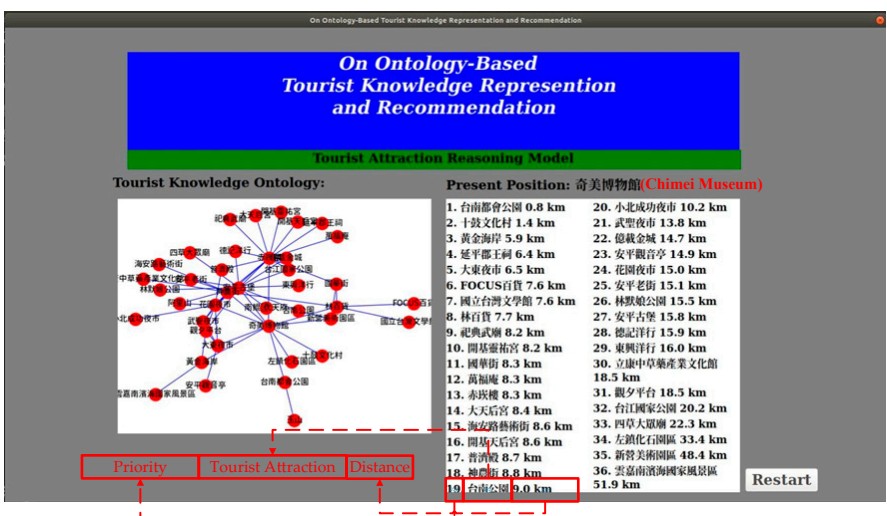

**Figure 21.** Results of tourist attraction reasoning model.

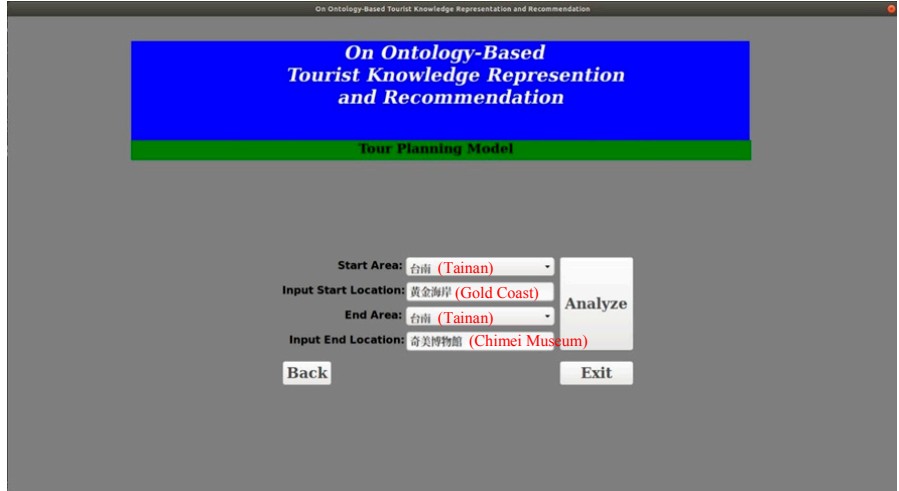

**Figure 22.** Input start node and end node.

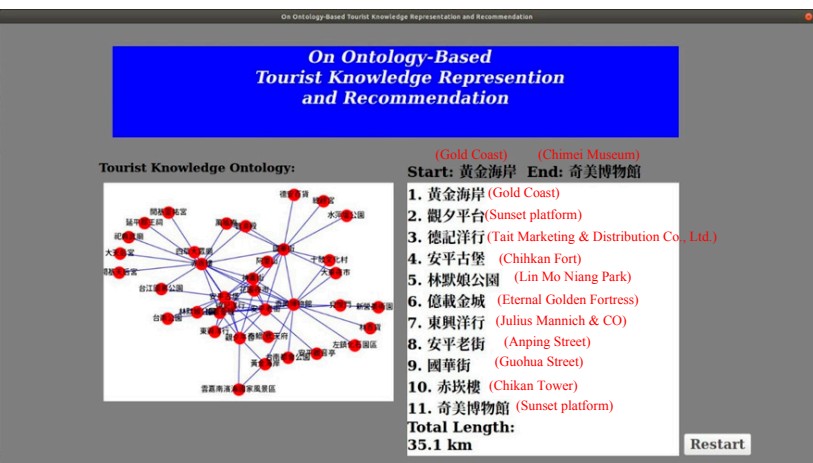

**Figure 23.** Results of tour planning model.

*4.3. System Evaluation*

In order to evaluate the effectiveness of this method and help travelers quickly look for popular attractions and tourism planning, this study conducted a satisfaction survey about this system after use. Currently, the user's satisfaction survey is commonly used in information system assessments to assess the efficacy of the system. The user's satisfaction survey in this study was extracted from the research by Li and Chang (2009) [29], Pai et al. (2013) [6], and Lin et al. (2017) [25]. The criteria of satisfactory assessment is based on the Likert scale, where 1 = very disagree, 3 = not determined, 5 = very agree.

The system evaluation of this study was conducted in control and experimental groups. The participation of control groups is mainly for evaluating the traditional search methods (ex:blog, BBS, search engine); the experimental groups participation is for the evaluation of the system. Then, based on the information from the participants of control and experimental groups experiences, t-test was used to understand what the significant difference between the traditional search methods and the research is. Therefore, 60 students were tested for the system evaluation, these students must have experience in using BBS (bulletin board system), Blog, and search engine to search for popular attractions and travel planning.

According to Table 4, the analysis result showed that the method of this study is able to quickly recommend popular attractions and travel planning, which is better than the traditional way of blog, BBS, search engine. However, this study is a short-term system assessment to help the user. A completed system evaluation is required to take 3 to 5 years, so the study advises that a long-term research is conducted in the future to improve the effectiveness of the system.

**Table 4.** Results of the performance evaluation of the proposed consumer demands analysis method.

| Questionnaire Item | Control Group | | Experimental Group | | T | *p* |
|---|---|---|---|---|---|---|
| | Mean | S.D. | Mean | S.D. | | |
| 1. The analytical range of the tourist attractions analysis is very wide. | 3.08 | 0.67 | 3.48 | 0.59 | −3.848 | ** 0.0001 |
| 2. The tourist attractions analysis reduced the time needed for tour planning. | 2.95 | 0.76 | 3.40 | 0.49 | −3.719 | ** 0.0002 |
| 3. The tourist attractions analysis can be used to analyze information about tourist attractions changes. | 3.05 | 0.76 | 3.43 | 0.53 | −3.223 | ** 0.001 |
| 4. The tourist attractions analysis does not require any training or expertise. | 3.15 | 0.63 | 3.48 | 0.59 | −3.250 | ** 0.0009 |
| 5. The data source for the tourist attractions analysis is reliable. | 3.03 | 0.63 | 3.38 | 0.55 | −3.708 | ** 0.0002 |
| 6. The analytical results of the tourist attractions analysis are objective. | 3.12 | 0.71 | 3.43 | 0.53 | −2.804 | ** 0.003 |
| 7. The tourist attractions analysis is effective. | 2.97 | 0.68 | 3.42 | 0.53 | −3.778 | ** 0.0001 |

\* *p* Value < 0.05 indicates significance at 0.05. \*\* *p* Value < 0.01 indicates significance at 0.01.

## 5. Conclusions and Further Work

The study applied ontology into the development of a tourist knowledge representation and recommendation mechanism. It is hoped that through the application of Internet, tourist knowledge can be effective managed and used. It may be able to turn the OTR on various blogs and web forums into valuable information. Travelers can extract helpful tourist attraction information to assist their traveling decision-making. In addition, the method may also be able to help tourism business to understand current popular tourist attractions. The tourism business may use the information as the reference to design their tourism products. It may improve their competitiveness in the tourism market. The primary results and contributions of this study are summarized as follows:

(1) Ontology-based tourist knowledge analysis: Based on the features of OTR and ontology, this study designed an ontology-based tourist knowledge representation and recommendation model to analyze OTR content. It is expected that this can help users to find knowledge to better plan tours.

(2) Ontology-based tourist knowledge representation and recommendation method: Based on the ontology-based tourist knowledge representation and recommendation model, the study developed a method for ontology-based tourist knowledge representation and recommendation. The method includes an OTR collection module, OTR content analysis module, tourist knowledge ontology construction module, and tourist attraction recommendation module.

(3) eWOM analysis mechanism: The study developed a method for ontology-based tourist knowledge representation and recommendation. Through the automatization of identifying and analyzing OTR, it is expected that this approach can provide the tourist knowledge that travelers need when planning their trips.

(4) This system evaluation of this study was conducted according to the questionnaires of Li and Chang (2009) [29], Pai et al. (2013) [6], and Lin et al. (2017) [25]. The results showed that the breadth search, the operation method, and the analysis results are better than the traditional ones. After an interview with the user, in their opinion, the advantage of this system is to find the link and idea for each tourist attraction, and to provide the users with new tourists.

(5) According to the research by Li et al. (2019) [30], text corpus-based tourism helps to observe and realize changes in the tourism market. OTR in this study is viewed as a kind of text corpus based tourism, which of the direction is confirmed to be correct, because the study by Gong, et al. (2018) [31] has the same opinion. However, the source information between Gong, et al. (2018) and this study is different. Gong, el al. (2018) is mainly for the tourist packages; this study is for OTR. In terms of data analysis, OTR collate the travelers' negative and positive appraisals, but tourist packages are from the enterprise's perspective. Therefore, the OTR data is more objective than the tourist packages.

Based on the proposed model and method in this study, the following future research directions are recommended:

(1) This study extracts tourist knowledge from OTR content, which is then used as the basis for developing a tourist attraction reasoning and tour planning method. Nevertheless, tourist attractions are not the only factor that travelers take into consideration when the tourists are planning their tour. They may also consider issues such as time, budget, age, and preferences. These factors can thus be included in future studies to provide better recommendations.

(2) In order to produce a tourist knowledge structure that meets the need of travelers, an ontology adapting and management mechanism can be considered in future research, as this could help to better understand and maintain the tourist knowledge structure. In addition, OTR content from tourist experts can also be collected to construct another knowledge structure. In this way, a multifaceted tourist knowledge management mechanism can be developed to provide travelers with a variety of recommendations.

(3) As far as the limitation of this study is concerned, the proposed methods may not be able to judge OTR effectively, since some OTR content is sponsored by businesses. While such sponsored content does not affect the design of the mechanism, it should be remembered that "garbage in, garbage out". This study hypothesized that all the OTR it collected was genuine content and not sponsored by businesses. Future studies could develop mechanisms that can filter sponsored OTR content out from genuine content, and this could improve the quality of the analyzed tourist knowledge.

(4) The study used tourist knowledge ontology to develop a recommendation mechanism for tourist attractions. This mechanism was primarily developed for the travelers. However, future studies could design mechanisms for enterprises to further raise the value of tourist knowledge.

(5) In this study, a short-term systematic assessment was done through the questionnaire, but a good system assessment is required to take three to four years. In the future, the system will proceed to evaluate continuously, which will be a basis function for the system improvement. In addition, the target participant in the future system evaluation will be the community, the industry, or the enterprise. And thus, the practicality and applicability of the system will be enlarged and broadened.

(6) Most of the OTR content only collates the appraisal of popular attractions, not the attributes of OTR authors, such as preferences, religious, beliefs, gender, and so on. The limitation of this study is that OTR author attributes cannot refer to the tourist attraction recommendation, because ORT author attributes cannot be analyzed. However, the user's operation record can be included and analyzed as a reference. The future research will make an effort to analyze the user attributes of the system and the inference results. And thus, this system analysis will meet the users' requirements more suitably.

**Author Contributions:** Conceptualization, M.-Y.P., D.-C.W., and C.-C.C.; Formal analysis, T.-H.H., and G.-Y.L.; Methodology, M.-Y.P.; Project administration, D.-C.W., and T.-H.H.; Software, T.-H.H., and G.-Y.L.; Writing—original draft, M.-Y.P., and C.-C.C.; Writing—review & editing, M.-Y.P.

**Funding:** This work was supported by Ministry of Science and Technology (MOST) of Taiwan (R.O.C) under Grants MOST 108-2218-E-020-003, 108-2221-E-034-015-MY2, 107-2221-E-006-017-MY2, 107-2218-E-006-055, 107-2221-E-218-024, and 107-2221-E-156-001-MY2.

**Conflicts of Interest:** The authors declare no conflict of interest.

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
