# Peer review of "On Ontology-Based Tourist Knowledge Representation and Recommendation"

_applsci, doi:10.3390/app9235097_

Round 1

Reviewer 1 Report

The review of the paper titled „On Ontology-Based Tourist Knowledge Representation and Recommendation”:

- In the abstract the authors could provide an information what the abbrev eWOM stands for, which is electronic word-of-mouth marketing

- Whereas the procedure of building the system has been extensively presented in the paper, the data used for its illustrative purposes have not. Actually, the authors could present some real-based examples of the use of their application (maybe it is advisable to translate labels on figures 19-23 into English) also supplying the information on how the data were collected to the system and by who (tourist themselves, through agencies, municipalities, etc.). The authors should give some descriptive statistics of the data collected by the system, so the Reader could verify their distribution in terms of the presented methods and measures used to analyse the data. It could be i.e. a case of a one commune, with attractions marked on the map and briefly discussed in the paper.

- Was the system tested on real tourist data? The authors could dwell on its possible links with tourist forums, social media, etc. to collect the data.

- In the concluding part the authors could be more precise in the way the system can be implemented and give positive feedback not only for tourist but also local communities, travel agencies, tourist agents, seeking for the most popular attractions or generating profits from the knowledge of travellers’ preferences.

- Adding additional tourist preferences into the system – let’s say - national heterogeneity or nationality (given the customs, religion, habits) – could present the reviews more tailored to the needs of particular tourists coming from different countries. The authors could regard to that idea in the concluding part.

- Finally, whether the system can be implemented for attractions located in different countries or in one country only? If the latter it is the limitation of the system.

Author Response

Reviewer1 Comments:

In the abstract the authors could provide an information what the abbrev eWOM stands for, which is electronic word-of-mouth marketing.

Reply:

We appreciate for Reviewer1 valuable suggestion. We already amend and add the abstract and keywords into the related content of eWOM.

Whereas the procedure of building the system has been extensively presented in the paper, the data used for its illustrative purposes have not. Actually, the authors could present some real-based examples of the use of their application (maybe it is advisable to translate labels on figures 19-23 into English) also supplying the information on how the data were collected to the system and by who (tourist themselves, through agencies, municipalities, etc.). The authors should give some descriptive statistics of the data collected by the system, so the Reader could verify their distribution in terms of the presented methods and measures used to analyse the data. It could be i.e. a case of a one commune, with attractions marked on the map and briefly discussed in the paper.

Reply:

Fingures 19-23 is amended. Please help us revise to the new content. Thanks for your help. Please refer to Fingers 19-23. As for the collation of OTR content, we use the study of Pai et al.for reference. In addition, we also collate the result of OTR content. We have an explanation in line 519-524. Thanks for your suggestion. Besides, we conduct the system analysis of sixty travelers in order to confirmed with the system’s feasibility and applicability. Please refer to Line 516-522.

Was the system tested on real tourist data? The authors could dwell on its possible links with tourist forums, social media, etc. to collect the data.

Reply:

Our system has been actually tested. We invite sixty travelers to evaluate the system. Please refer to Line 528-548. In additon, we conduct the OTR contest collation and analysis, based on the research method of Pai et al. Please refer to line 517-522. To void the readers’ confusion with the study purpose, we only simply explain the source information of OTR, the quantity of ORT content collation and how to find the popular attractions through OTR . We hope that the reader focus on the development and method of travel recommendation. Thus, we only have an simple explanation for the quantity of OTR collation.

- In the concluding part the authors could be more precise in the way the system can be implemented and give positive feedback not only for tourist but also local communities, travel agencies, tourist agents, seeking for the most popular attractions or generating profits from the knowledge of travellers’ preferences.

Reply:

We believe that our study will be beneficial for the local communities, travel agencies, tourist agents. The effectiveness of the system evaluation can be a study. However, if we explore the topic, our study content will be too large. Thus, we only mention some of the information in conclusion. Please refer to Line 573-585. The study do the analysis of the users satisfaction survey. The system evaluation from travelers can help us to find the problem or disadvantage, and then can be as a reference for improving our system. Please refer to Line 527-548.

- Adding additional tourist preferences into the system – let’s say - national heterogeneity or nationality (given the customs, religion, habits) – could present the reviews more tailored to the needs of particular tourists coming from different countries. The authors could regard to that idea in the concluding part.

Reply:

Thanks for Reviewe1’s suggestion. We already amend our content. Please refer to line589-593 and 610-622, mark in read.

- Finally, whether the system can be implemented for attractions located in different countries or in one country only? If the latter it is the limitation of the system.

Reply:

Please refer to figer1. Of our study. We have developed many different application analysis techniques for eWOM and OTR. These analysis technologies have been modularized and can be replaced. For example, we only need to change the language fo OTR collection and analysis method to English; and then we can search for the attractions of English-speaker countries.

Figure 1. OTR & eWOM management framework.

Reviewer 2 Report

I would suggest to formulate an abstract in a standard format, to explain the topic, then the aim and object of the research, the methods of research used, the results and the recommendations. The abstract in its current form overlaps from the “Introduction” section which I do not consider appropriate.

In the “Introduction” section I recommend to explain more and more deeply the motivation of the authors for the realized research.

Line 132 does not indicate the source of the scheme, it is unclear to me whether the scheme is the work of the authors, a compilation of the cited works (although line 128 states "Pai" but does not refer to the source ") or unquoted schema.

In line 147, better explain 'Consumer demands analysis' and its link to the source cited (25).

In line 177 there are, in my view, two partially unreadable schemes with inappropriate formatted text that partially overlaps. I strongly recommend that you rework the journal rules for image attachments.

The whole chapter 2.2-2.3 does not refer to any relevant source on which the analysis is based, I recommend adding references to similar research that would codify the authors' procedure. Respectively for their own published works, if the analyzes have no starting point in the state of knowledge of the discipline.

Line 258, add reference to 'Chen (2010)'.

Line 298, in the schema are unreadable objects, reworked according to journal rules.

Line 511, 513, 515, 517, 519 the same, see above.

In line 519, in addition to the confusing scheme, I recommend adding a legend for text in foreign language.

Schemes 19-23 should be moved to the “attachments” section if the journal rules allow.

Summary:

Insufficient elaboration of the current state of knowledge in terms of quantity and timeliness of sources, I recommend adding to the text references to relevant studies, similar researches and analyzes, as well as to clearly define and cite the theoretical fundamentals on which the study is based.

I recommend to elaborate the paper into the standard IMRaD structure, which should correspond not only to the content of the paper but also to the structure of its chapters.

I would point out the disproportionate scope of the paper (contribution is excessively long). I recommend either moving the picture attachments (and reworking them graphically) to a part of the annex, splitting the paper into two studies, or reducing it to a range that corresponds to the standard scope of similar studies.

Last but not least, I recommend the authors to extend the “Conclusions” section by linking them to theoretical bases and relevant studies of a similar nature, followed by comparing the results (if possible).

Author Response

Reply Letter to Reviewer2 Comments (applsci-620633)

I would suggest to formulate an abstract in a standard format, to explain the topic, then the aim and object of the research, the methods of research used, the results and the recommendations. The abstract in its current form overlaps from the “Introduction” section which I do not consider appropriate.

Reply:

Thanks for Reviewer2 suggestion. We already amend the content. Please refer to the new content. Please refer to line 18-34.

In the “Introduction” section, I recommend to explain more and more deeply the motivation of the authors for the realized research.

Reply:

We already amend and add the related research. Please refer to the line 65-68.

Line 132 does not indicate the source of the scheme, it is unclear to me whether the scheme is the work of the authors, a compilation of the cited works (although line 128 states "Pai" but does not refer to the source ") or unquoted schema.

Reply:

Literature information has been included, which will help our content more completed. Thanks for your suggestion. Please refer to the line 132.

In line 147, better explain 'Consumer demands analysis' and its link to the source cited (25).

Reply:

We already amend and re-explain “consumer demands analysis. Please refer to line 151-155.

In line 177 there are, in my view, two partially unreadable schemes with inappropriate formatted text that partially overlaps. I strongly recommend that you rework the journal rules for image attachments.

Reply:

Thanks for Reviewer2 suggestion. We already amend the content. Please refer to the new content. Please refer to line 178-191.

We already amend. Please refer to Line 555, marked in read. The whole chapter 2.2-2.3 does not refer to any relevant source on which the analysis is based, I recommend adding references to similar research that would codify the authors' procedure. Respectively for their own published works, if the analyzes have no starting point in the state of knowledge of the discipline.

Reply:

Thanks for your suggestion. The reference for section 2.2-2.3 is increased. Please refer to line 230-233.

Line 258, add reference to 'Chen (2010)'.

Reply:

Thanks for your suggestion. We already add reference. Please refer to line 268.

Line 298, in the schema are unreadable objects, reworked according to journal rules.

Reply:

The related content is amended as line33, marked in read. Please refer to 302-310.

Line 511, 513, 515, 517, 519 the same, see above.

Reply:

Thanks for Reviewer2 suggestion. We already amend the content. Please refer to the new content. Please refer to line 493-505.

In line 519, in addition to the confusing scheme, I recommend adding a legend for text in foreign language.

Reply:

The related reference has been added. Please refer to finger 17-19.

Schemes 19-23 should be moved to the “attachments” section if the journal rules allow.

Reply:

We already move. Please refer to finger 17-19, thanks for Reviewer2 suggestion.

-------------------------------------------------------------------

Summary:

Insufficient elaboration of the current state of knowledge in terms of quantity and timeliness of sources, I recommend adding to the text references to relevant studies, similar researches and analyzes, as well as to clearly define and cite the theoretical fundamentals on which the study is based.

Reply:

Please refer to line517-522 and 528-548, marked in read. We have an explanation about the related study. Thanks for your suggestion. The study purpose is to conduct the analysis ORT content, based on the research method of Pai et al (2013) To void the reader’s confusion, we cannot mention too many content of Pai et al’s research method. We are worry that the reader focus one the development of OTR analysis, not on the tourist knowledge representation and recommendation

I recommend toelaborate the paper into the standard IMRaD structure, which should correspond not only to the content of the paper but also to the structure of its chapters.

I would point out the disproportionate scope of the paper (contribution is excessively long). I recommend either moving the picture attachments (and reworking them graphically) to a part of the annex, splitting the paper into two studies, or reducing it to a range that corresponds to the standard scope of similar studies.

Reply:

Our study refer to the study structure by Pai et al (2013), Chen and Chen (2012) and Lin et al(2017). We believe the study structure meet the standard of IMRad. We already amend the finger and will discuss with the journal editor about where the finger can be shown.

Last but not least, I recommend the authors to extend the “Conclusions” section by linking them to theoretical bases and relevant studies of a similar nature, followed by comparing the results (if possible).

Reply:

Thanks for your suggestion. We will add some of discussion in conclusion, which make the content more completed.

Round 2

Reviewer 2 Report

Technicalities:

Please see your Cover letter, the part, where you responded on my recommendations, specifically:

"In the “Introduction” section, I recommend to explain more and more deeply the motivation of the authors for the realized research.

Your Reply:

We already amend and add the related research. Please refer to the line 65-68."...

My comment on that:

I did not see nothing related to my recommendation.

----

Once again, please see your Cover letter, the part, where you responded on my recommendations, specifically:

"I recommend to elaborate the paper into the standard IMRaD structure, which should correspond not only to the content of the paper but also to the structure of its chapters.

I would point out the disproportionate scope of the paper (contribution is excessively long). I recommend either moving the picture attachments (and reworking them graphically) to a part of the annex, splitting the paper into two studies, or reducing it to a range that corresponds to the standard scope of similar studies.

Your Reply:

Our study refer to the study structure by Pai et al (2013), Chen and Chen (2012) and Lin et al(2017). We believe the study structure meet the standard of IMRad. We already amend the finger and will discuss with the journal editor about where the finger can be shown."

My comment on that:

I leave the journal accepting or not accepting this structure.

----

The rest can be considered fulfilled, however, adjustments are mostly made using "creativity" as opposed to the complex and laborious adjustments I would expect.

I leave the question of formal consistency with journal requirements to the editor.

----

Sincerely...

Author Response

Reply Letter to Reviewer2 Comments (applsci-620633)

Thanks for your feedback. Please refer to the below for our reply.

Reviewer2 Comments:

Comment 1:

Please see your Cover letter, the part, where you responded on my recommendations, specifically:

"In the “Introduction” section, I recommend to explain more and more deeply the motivation of the authors for the realized research.

Your Reply:

We already amend and add the related research. Please refer to the line 65-68."...

My comment on that:

I did not see nothing related to my recommendation.

Reply:

Thanks for your suggestion. As for “motivation of the authors for the realized research”, please refer to Line 81-90.

In addition, we also add the content about the travelers’ searching difficulties in looking for travel information. Please refer to Line 81-90.

Finally, according to Reviewer 1’s suggestion, we add the context about the comparison with similar studies in final conclusion. Please refer to Line 578-585.

Comment 2:

----

Once again, please see your Cover letter, the part, where you responded on my recommendations, specifically:

"I recommend to elaborate the paper into the standard IMRaD structure, which should correspond not only to the content of the paper but also to the structure of its chapters.

I would point out the disproportionate scope of the paper (contribution is excessively long). I recommend either moving the picture attachments (and reworking them graphically) to a part of the annex, splitting the paper into two studies, or reducing it to a range that corresponds to the standard scope of similar studies.

Your Reply:

Our study refer to the study structure by Pai et al (2013), Chen and Chen (2012) and Lin et al(2017). We believe the study structure meet the standard of IMRad. We already amend the finger and will discuss with the journal editor about where the finger can be shown."

My comment on that:

I leave the journal accepting or not accepting this structure.

----

Reply:

Thanks for your suggestion. We already amend Section 2 “ Ontology‐Based Tourist Knowledge Representation and Recommendation Model Design” is amended to “OTR and Ontology Characteristics Analysis”. Please refer to Line 122.

IMRaD’s study structure is Introduction, Methods, Results and Discussion. However, our main study structure is 1.Introduction. OTR and Ontology Characteristics Analysis.  3. Ontology-Based Tourist Knowledge Representation and Recommendation Method Development.  4. Prototype Implementation and System Evaluation.  5. Conclusions and Further Work.

We have the further and detailed explanation about the relationship between IMRaD and our research, showing as below.

Introduction: this mainly explains our research background and motivation. And thus we can confirm that the tourism recommendation through OTR is valuable and this issue is worth developing into a research topic.

Methods:The introduction of the study confirms the study value and development direction is OTR. Thus, the research do OTR and ontology characteristics analysis and then help us realize the data characteristics of OTR and ontology. The study can carry on the system design based on ORT and ontology characteristics. With the explanation of our data characteristics, we can prove that our system structure and design is reasonable.

After understanding OTR and ontology characteristic, we can carry on Ontology-Based Tourist Knowledge Representation and Recommendation Method Development.

Results: In this study, chapter 4 of Prototype Implementation and System Evaluation shows the system development results. Further, we explain the system usage through the questionnaire.

Discussion: finally, we have an discussion according to our research results.

In summary, our research is based on IMRaD. In the past, we also use the same writing methods, suchas Pai et al.,(2013a), Pai et al.,(2013b), Lin et al.,(2017), Chen et al., (2012), Chen et al., (2010). IMRaD is a standard; our past research also met the standard of IMRaD. Thus, we can publish the research result in SCI journals, which is as famous as applied science. Take an example of Electronic Markets, Decision Support Systems, Expert Systems with Applications and Knowledge-Based Systems. Now, we use the same method to write the research content. Therefore, the research structure is in accordance with the rules of IMRaD.

References

-Y. Lin, S.-Y. Liaw, C.-C. Chen, M.-Y. Pai, and Y.-M. Chen, A computer-based approach for analyzing consumer demands in electronic word-of-mouth, Electronic Markets, 2017, 27(3), 225-242. -Y. Pai, H.-C. Chu, S.-C. Wang, and Y.-M. Chen, Electronic word of mouth analysis for service experience, Expert Systems with Applications, 2013a, 40(6), 1993-2006. -Y. Pai, H.-C. Chu, S.-C. Wang, and Y.-M. Chen, Ontology-based SWOT analysis method for electronic word-of-mouth, Knowledge-Based Systems, 2013b, 50, 134-150. -J. Chenand Y.-M. Chen, Knowledge Evolution Course Discovery in a Professional Virtual Community, Knowledge-Based Systems, Vo. 33, pp. 1-28, 2012. (SCI) -J. Chen, Development of a Method for Ontology-Based Empirical Knowledge Representation and Reasoning, Decision Support Systems, Vol. 50, No. 1, pp. 1-20, 2010. (SSCI/SCI)
